# Endovascular Treatment of Chronic Subdural Hematomas through Embolization: A Pilot Study with a Non-Adhesive Liquid Embolic Agent of Minimal Viscosity (Squid)

**DOI:** 10.3390/jcm10194436

**Published:** 2021-09-27

**Authors:** Andrey Petrov, Arkady Ivanov, Larisa Rozhchenko, Anna Petrova, Pervinder Bhogal, Alexandru Cimpoca, Hans Henkes

**Affiliations:** 1Vascular Neurosurgery Department, Polenov Neurosurgical Research Institute, Branch of Almazov National Medical Research Centre, 191014 St. Petersburg, Russia; doctorpetrovandrey@gmail.com (A.P.); arkady.neuro@gmail.com (A.I.); rozhch@mail.ru (L.R.); petrovaanna2803@gmail.com (A.P.); 2Interventional Neuroradiology Department, The Royal London Hospital, Barts NHS Trust, London E1 1BB, UK; bhogalweb@gmail.com; 3Neuroradiological Clinic, Klinikum Stuttgart, 70174 Stuttgart, Germany; a.cimpoca@klinikum-stuttgart.de; 4Medical Faculty, University Duisburg-Essen, 45122 Essen, Germany

**Keywords:** endovascular embolization, chronic subdural hematoma, Squid, middle meningeal artery, ethylene-vinyl alcohol copolymer

## Abstract

Objective: Endovascular embolization using non-adhesive agents (e.g., ethylene vinyl alcohol copolymer with suspended micronized tantalum dissolved in dimethyl sulfoxide; Squid, Balt Extrusion) is an established treatment of brain arteriovenous malformations, dural arteriovenous fistulas, and hypervascular neoplasms. Middle meningeal artery (MMA) embolization is a relatively new concept for treating chronic subdural hematomas (CSDH). This study aimed to evaluate the safety and effectiveness of the use of Squid in the endovascular treatment of CSDH. Methods: Embolization was offered to patients with CSDH with minimal or moderate neurological deficits and patients who had previously undergone open surgery to evacuate their CSDH without a significant effect. Distal catheterization of the MMA was followed by embolization of the hematoma capsule with Squid 12 or Squid 18. Safety endpoints were ischemic or hemorrhagic stroke and any other adverse event of the endovascular procedure. Efficacy endpoints were the feasibility of the intended procedure and a ≥ 50% reduction of the maximum depth of the CSDH confirmed by follow-up computed tomography (CT) after >3 months. Results: Between November 2019 and July 2021, 10 patients (3 female and 7 male, age range 42–89 years) were enrolled. Five patients had bilateral hematomas, and five patients had previously been operated on with no significant effect and recurrent hematoma formation. The attempted embolization was technically possible in all patients. No technical or clinical complication was encountered. During a post-procedural follow-up (median 90 days), 10 patients improved clinically. A complete resolution of the CSDH was observed in 10 patients. The clinical condition of all enrolled patients during the so-far last contact was rated mRS 0 or 1. Conclusion: A distal catheterization of the MMA for the endovascular embolization of CSDH with Squid allowed for the devascularization of the MMA and the dependent vessels of the hematoma capsule. This procedure resulted in a partial or complete resolution of the CSDH. Procedural complications were not encountered.

## 1. Introduction

The rupture of bridging intracranial veins can cause a hemorrhage in between the dura mater and the arachnoid (the subdural space). The term “chronic” subdural hematoma (CSDH) refers to a clinical onset 3 weeks after the presumed causing event. CSDHs occur associated with various predisposing conditions, including minor blunt head trauma, brain atrophy, arterial hypertension, cerebrovascular atherosclerosis, excessive alcohol consumption, diabetes mellitus, use of medicinal anticoagulation or antiaggregation, and spinal leakage of cerebrospinal fluid. The first step in the pathophysiology of CSDH formation is the separation of the dural border cell layer, followed by inflammation and proliferation of dural border cells. The invasion of macrophages and granulation tissue formation results in a surrounding membrane. This membrane releases angioneogenetic factors, inducing the angioneogenetic formation of new vessels (“neovascularization”). A possible explanation for CSDH growth through recurrent hemorrhages is microvascular bleeding inside the membrane. The neovascularization originates from distal branches of the regional artery (i.e., in the far majority of the cases, the MMA) [1]. Embolization of the MMA necessitates the injection of the vessel with a contrast medium, and contrast staining is often seen on the postoperative CT scans, which indirectly confirms the connection between the MMA, the capsule, and the content of the CSDH [2,3]. It is believed that embolization of the MMA interferes with the final step of the pathophysiological cascade of CSDH formation [4,5,6].

The incidence of CSDHs increases with age, with the majority of patients being >65 years old. The clinical signs and symptoms are usually progressive or fluctuate with headaches, confusion, dementia, incontinence, epileptic seizures, and focal neurological deficits. Large-volume CSDH may cause brain herniation and death.

With an incidence of 2–20 per 100,000 a year in the general population and 50–60 cases per 100,000 persons >70 years of age a year, CSDH is one of the most frequent disorders in neurosurgery [7]. In symptomatic patients, surgical evacuation of the CSDH is standard practice. Burr hole trephination, twist drill craniotomy, and craniotomy yield similar safety and efficacy results [8,9]. In 10–30% of the operated patients, recurrence of the CSDH will occur [10]. Craniotomy (instead of burr hole), postoperative blood in the operative field, and thrombocytopenia are associated with an increased likelihood of postoperative CSDH recurrence without embolization [11]. In bilateral CSDHs, the recurrence rate is even higher than in unilateral CSDH [12,13].

The formation of a membrane around the CDSH is a crucial aspect of the pathophysiology of this disorder. Way beyond the shear rupture of superficial veins, complex processes including angiogenesis, fibrinolysis, and inflammation together maintain the formation of the said membrane [14,15].

Tanaka et al. (1998) described the angiographic appearance of middle meningeal arteries (MMA) supplying CSDHs. They found dilated MMAs and abnormal vascular networks at the site of the CSDH membranes [16]. Pouvelle et al. (2020) confirmed the increased caliber of the MMAs adjacent to CSDHs [17]. Ishihara et al., in 2007, embolized the MMAs with the dependent angioneogenetic membrane vasculature of seven patients with recurrent CSDH using 20% nBCA diluted with Lipiodol (Guerbet) and reported good results [18]. Ban et al., in 2018, compared patients with CSDH who underwent conservative observation (*n* = 67), surgery (*n* = 469), or embolization (*n* = 72) with polyvinyl alcohol particles (150–250 μm) and achieved the best results in the group with endovascular treatment [19].

By 19 July 2021, Pubmed Central shows 155 hits for the search items “chronic subdural hematoma” and “embolization.” Despite this widespread interest, a randomized controlled trial comparing conservative management, open surgery, and MMA embolization for the treatment of CSDH is pending [20]. One of the open questions in this context concerns the embolic agent to be used. It is still controversial whether nBCA/Lipiodol, polyvinyl alcohol particles, or non-adhesive agents (Onyx, Medtronic, Dublin, Ireland; Squid, Montmorency, France; PHIL, MicroVention, Tustin CA, USA) are equally safe and effective for this indication.

Our goal was to (1) evaluate Squid as an embolic agent for the MMA embolization of patients with ipsilateral untreated or recurrent CSDH, and (2) provide an updated review and summary of the current literature on this topic.

We present the first series of MMA embolizations for CSDH treatment with the exclusive usage of Squid (Balt Extrusion). Squid was chosen instead of Onyx (Medtronic) because a low-viscosity variant is available [21].

## 2. Materials and Methods

### 2.1. Adherence to Ethical Standards

Before the initiation of the study, the responsible “Local Ethics Committee of the National Medical Research Almazov Centre” was consulted. Approval of the study was granted. The study was conducted in accordance with the World Medical Association’s “Declaration of Helsinki” in the 2013 version. All patients were informed about the treatment, the associated chances and potential risks, and the nature of the study. All patients declared informed consent in written form.

### 2.2. Inclusion and Exclusion Criteria

Inclusion criteria for possible participation in this study were defined as follows:-Patient with CSDH, confirmed by non-contrast computed tomography (NCCT) or MRI.-Patient being asymptomatic or symptomatic, without clinical signs of acutely increased ICP.-Patient without previous surgical treatment or.-Patient with recurrence after previous surgical treatment; no defined time interval between surgery and embolization was applied; a minimum depth of 10 mm was considered necessary to justify embolization.-Patient able to understand the purpose of the study.-Patient able to tolerate the endovascular procedure.-Patient > 18 years old.-Patient is not pregnant.

Exclusion criteria were:-Patient with symptomatic CSDH, confirmed by NCCT or MRI, with clinical signs of acutely increased ICP (e.g., impaired consciousness, vomiting without nausea, papilledema).-Patient not able to understand the purpose of the study.-Patient not able to tolerate the endovascular procedure (e.g., severe allergy against contrast medium, severe renal insufficiency).-Patient < 18 years old.-Patient is pregnant.-Visible anastomosis between the MMA and the ophthalmic artery.

### 2.3. CSDH Volume Measurement

CSDH volume was assessed based on pre-embolization non-contrast head CT scans using OsiriX software (Osirix for Mac, version 11.0) which allows the measurement of defined volumes with a manual definition of the region of interest (Figure 1).

### 2.4. Endovascular Treatment Strategy and Technique

The strategy of the endovascular treatment of CSDH was oriented towards the following goals:-Direct embolization of the distal MMA branches in a single session.-Avoidance of reflux into proximal segments of the MMA in order to avoid the dissemination of Squid through dangerous anastomoses.-Termination of blood supply to the vessels of the CSDH capsule through occlusion of the meningeal arteries and of the vessels of the capsule of the CSDH by non-adhesive embolizing material of low viscosity in order to prevent recurrence and increase the hematoma volume.-Penetration of the non-adhesive embolizing agent through collaterals to distal branches of the opposite MMA, preventing blood supply of the CSDH capsule from the opposite side.-Acceleration of the processes of hematoma absorption and decompression of the adjacent brain.

A 6F guiding catheter was introduced into the external carotid artery ipsilateral to the CSDH, and a diagnostic biplane DSA was obtained. An anastomosis between the MMA and the ophthalmic artery would have disqualified the patient from the study. The anterior and posterior branches of the MMA have been identified angiographically. The petrous branch of the MMA and vessels from which EC-IC collaterals exist were not embolized [2,22].

Under roadmap guidance, the MMA was then catheterized with an Apollo (Medtronic) or Sonic (Balt Extrusion) microcatheter.

Microangiography of the MMA revealed a hypervascular network with the typical “cotton-wool” pattern in all patients. The supply came from the distal branches of the MMA ipsilateral to the hematoma, as an angiographic correlate of the process of hematoma capsule formation described earlier (Figure 2).

Ten patients underwent endovascular embolization of the concerning MMA. 

In the unilateral group, 5 patients underwent embolization of the frontal branch of the middle meningeal artery and the capsule vessels on one side (at the side of the hemtoma).

In the bilateral group, comprising 5 patients with bilateral CSDHs, sequential embolization of the MMAs on both sides was performed.

On average, one vial of Squid 12 and one or two vials of Squid 18 per side were used.

The average injected volume was:

Squid 12 (mean 0.92 mL, median ± SD 0.9 ± 0.48 mL) per side.

Squid 18 (mean 2.1 mL, median ± SD 1.7 ± 0.79 mL) per side.

Bilateral CSDH (5/10 patients) required bilateral MMA embolization. Therefore 15 embolizations were performed in 10 patients.

The technical details of the procedure are as follows:-Far distal catheterization of the frontal and parietal branch of the middle meningeal artery, almost to the level where the outer diameter of the microcatheter coincides with the inner lumen of the artery.-DSA with contrast medium injection via the microcatheter in the MMA, demonstrating the “cotton wool” areas of neovascularization.-From this wedged position, the first portion of Squid 12 is injected to obliterate all MMA branches distal to the catheter tip, including the CSDH capsule’s angioneogenic vessels.

Squid12 is replaced by Squid18, which forms a durable cork at the level of the detachable part of the microcatheter and simultaneously pushes previously injected portions of the embolizing agent of lower viscosity distally along the MMA into collaterals on the opposite side and further into the CSDH capsular vessels.

At this point, it is recommended to switch again to Squid 12 to achieve maximum penetration of the embolizing agent into the vessels of the capsule, filling and thus destroying this vascular network almost all the way to the draining veins. This leads to the simultaneous shutdown of the neomembrane and all its derivative arterioles, microcapillaries, and venules.

Embolization should be limited to the frontal and parietal branches of the MMA, avoiding the ophthalmic artery anastomosis, the petrosal branch, and the foraminal segment.

### 2.5. Follow-Up Examinations

Patients underwent scheduled NCCT examinations 1 day, 1 week, 1 month, 3 months, and, as much as possible, 6 months after the endovascular treatment unless otherwise mandated by the clinical circumstances. NCCT follow-ups were part of the study to rule out or recognize possible complications of the embolization treatment. DSA follow-up between month 3 and month 6 after the embolization was offered to all patients.

### 2.6. Study Endpoints

#### 2.6.1. Primary Endpoint (Safety Endpoint)

Procedural complications were used as primary endpoints. Primary endpoints were considered if patients developed an ischemic stroke, intracranial hemorrhage, cranial nerve palsy, a visual disturbance, an epileptic seizure, or any complication related to endovascular access (e.g., puncture site hematoma requiring surgical intervention, ICA, or CCA dissection).

#### 2.6.2. Secondary Endpoint (Efficacy Endpoint)

The secondary endpoint was reached if the follow-up NCCT demonstrated a ≥50% reduction of the CSDH volume without surgery.

## 3. Results

### 3.1. Patient Population

Between November 2019 and July 2021, endovascular treatment was offered to 12 patients and accepted by 10 patients (7 male) with CSDH, following the study’s inclusion and exclusion criteria. The median age of the patients was 66 (range 42–89 years old). Only seven patients reported head injuries either as a result of road accidents or falls. No patient reported orthostatic headache as a symptom of idiopathic intracranial hypotension due to spinal CSF leakage. Unilateral CSDHs were seen in five patients, and in five patients, bilateral CSDHs were diagnosed. No patient was receiving anticoagulants at the time of the injury. Five patients from this series had previously undergone neurosurgical operations for CSDH. One of them underwent bilateral drainage of CSDHs via burr holes. Four patients underwent resection trephinations with the evacuation of their hematomas.

In patient #5 with bilateral hematomas, the evacuation of the hematoma on the right-hand side led to a decrease in hematoma size.

In patient #6, NCCT demonstrated a complete restoration of the volume of the CSDH after surgical removal with an occurrence of motor aphasia and hemiparesis (rated as NHISS 4).

Out of five patients who were not operated on, two patients (one with bilateral and one with unilateral hematomas) did not have any neurological symptoms. Three other patients had neurological deficits:-A 70-year-old patient with left-handed CSDH and a midline shift of 12 mm, motor aphasia, and right-hand hemiparesis.-A 42-year-old patient with bilateral CSDH that progressively increased during 1.5 months from 54 mL to 69 mL on the right-hand side and from 68 mL to 83 mL on the left-hand side, causing significant compression of both hemispheres and neurological deterioration.-An 89-year-old patient with bilateral CSDH (no midline shift). There was no apparent connection with an antecedent head injury. However, the patient’s relatives reported that the patient did periodically fall. At the time of admission, the patient presented tetraparesis and a decreased level of consciousness.

Neurosurgeons considered all three patients (from the non-operated group) candidates for open surgery to remove the CSDH. However, after an interdisciplinary discussion, it was decided to perform embolization of the MMA first and then reconsider the need for open surgery. All three patients improved clinically within the first week after the embolization.

### 3.2. CSDH Volumes

The volumes of the 15 CSDHs were measured to be between 9 and 169 mL (mean 63.33 mL, median ± SD 65 ± 40.40 mL). The details of every patient are summarized in Table 1.

### 3.3. Follow-Up Examinations

All patients underwent at least one follow-up NCCT examination. Table 1 shows the number of days between the embolization and the follow-up NCCT and the concerning volume of the CSDH. So far, seven DSA examinations have been carried out, and they confirmed the permanent occlusion of all embolized MMAs.

#### 3.3.1. Safety Endpoints

There were no thromboembolic or hemorrhagic complications or technical failures of the attempted embolization of the MMA. Thus, no patient met the primary endpoint.

#### 3.3.2. Efficacy Endpoint

In 10/10 patients (100%), follow-up NCCT demonstrated a >50% volume reduction of the treated CSDH without intervening surgery. The NCCTs of a representative patient are shown in Figure 3.

### 3.4. Illustrative Cases

Patient #4 underwent surgical evacuation of a chronic subdural hematoma in another hospital and was admitted to our institution with a recurrent hematoma. NCCT revealed a complete reaccumulation of the hematoma volume and aggravation of the neurological deficit in the form of hemiparesis and motor aphasia. After embolization, the patient was in a stable condition with regression of the neurological deficit at the time of discharge. The NCCT scan after 2 months showed a decrease in hematoma volume. NCCT after 4 months showed the hematoma regressed and the brain expanded (Figure 4).

Patient #3 had sustained a head injury 30 days earlier and suffered from severe headaches and nausea since then. An NCCT revealed a growth of the bilateral CSDHs due to recent hemorrhages in the hematoma capsule. Surgical evacuation of CSDHs was proposed to the patient. The patient, however, refused the operation, and he was admitted to our hospital for endovascular treatment (Figure 5).

## 4. Discussion

CSDH is a frequent disease of elderly patients; however, only 6/10 patients were over 65 years old in our series of patients. There was a male predominance (7:3), and 7 out of 10 patients had sustained previous head trauma. Our patients did not receive antithrombotic or anticoagulant therapy. A peculiarity of our patients was that five had previously been operated upon but without any radiological improvement. Three were potential candidates for surgery in case of neurological deterioration.

Historically, the pathogenesis of CSDH has never been attributed to trauma alone. In 1857, based on postmortem studies, Virchow suggested that organized exudates accumulate in the subdural space due to a generalized inflammatory process [23]. He coined the term “internal hemorrhagic pachymeningitis.” Subsequent studies have shown that blood in the subdural space provokes a non-specific inflammatory response after head trauma and, eventually, leads to the formation of a vascular neomembrane, which is responsible for recurrent microhemorrhages. They are viewed as the trigger for the growth of the hematoma. Brain atrophy and a decrease in brain volume in older people create the possibility of a more significant displacement of the brain by trauma and an increase in the subdural space, predisposing them to the accumulation of blood in it. A positive correlation has been found between the frequency of CSDH and the severity of brain atrophy. In an atrophied brain, the absence of tamponade from the brain’s surface to the dura mater contributes to the development and expansion of the hematoma. CSDH can cause a long-term compression of the brain hemispheres, and the reduced elasticity of the brain of an older person prevents the brain from spreading after surgery and causes recurrent hematomas after a successful evacuation of CSDH.

Blood products mixed with subdural fluid cause a foreign body reaction, accompanied by the formation of a neomembrane. This membrane is formed due to an inflammatory response to blood in the subdural space. Fibroblast proliferation occurs, and angiogenic factors are released, leading to neovascularization of the membrane that contains an immature vascular network on the inner aspect of the dura. After the initial formation of the CSDH secondary to rupture of the bridging veins, repeated microhemorrhages from the fragile newly formed vessels of the neomembrane surrounding the CSDH are responsible for its continued growth and recurrence. Considerable increases in VEGF levels within the neomembrane have been previously reported [24]. In response to these abnormal levels of angiogenic factors, the neovasculature of the membrane and CSDH is kept in an immature and ‘leaky’ state, allowing plasma extravasation and continual growth of the CSDH. Selective angiography reveals an extensive, newly formed, irregular capillary network (“cotton wool cloud” or “cotton wool-like staining” [25] that penetrates the dura mater and is connected with the branches of the MMA. This angioneogenetic capillary network with increased blood flow through hypertrophied MMAs into the capillaries leads to recurrent microhemorrhages from the neomembrane. Taking anticoagulants or antiplatelet drugs could enhance this process. Histological results of the CSDH multilayer membrane, published by other groups, revealed

-Giant capillaries and macrophage infiltration in the outer layer and the inner layer.-Tiny newly formed capillaries with highly permeable endothelial gap junctions.-Endothelial cells expressing high levels of vascular endothelial growth factor (VEGF) and PEGF (placental endothelial growth factor).-Proliferating fibroblasts forming fibrous granulation tissue with collagen deposition.-Chronic lymphoplasmacytic and histiocytic inflammation.-Macrophages containing hemosiderin.

Despite also not being the focus of this paper, some surgical aspects of the management of CSDH are essential for a better understanding of the treatment issues of these patients. Several concise summaries of the development of state-of-the-art surgical CSDH treatments are recommended [26,27]. 

Perioperative MMA embolization: The combined performance of MMA embolization and burr-hole craniotomy or Subdural Evacuation Port System (SEPS) insertion is an interesting concept [28]. Ng et al. (2019) were able to show that a PVA embolization of the MMA after surgical evacuation results in faster absorption of the CSDH [29]. Schwarz et al. (2021) operated 44 CSDH and embolized the MMAs “soon after a surgical evacuation.” A >50% CSDH volume resolution was achieved in 40/44 CSDHs, and only 2 (4.5%) required a second operation within one year [30]. Nakagawa et al. (2019) embolized the MMAs of 20 patients with nBCA after a second postoperative recurrence of their CSDHs and performed a third burr-hole surgery after the embolization. None of these patients experienced a third recurrence [13]. For the treatment of organized CSDHs, a combination of preoperative MMA embolization followed by a small craniotomy has been recommended [31].

Medications: Several clinical trials and retrospective studies evaluated drugs in their efficacy on the development of CSDH [32]. Most of them so far have failed. Perindopril, for instance, an angiotensin-converting enzyme inhibitor, had no effect [33]. Tranexamic acid is an antifibrinolytic drug, which showed some benefit in selected patients with CSDH [34]. Data beyond this anecdotal report are pending.

Medicinal antiaggregation and anticoagulation are suspected of leveraging the growth of CSDHs but may not affect the frequency of postoperative recurrence [12]. Rajah et al. (2020) observed that the use of antiplatelet medication after the embolization correlated with failed absorption of the CSDH [35]. Yajima et al. (2020) found an increased rate of postoperative CSDH recurrence in patients under anticoagulation [36].

With regards to the embolization of the MMA as a treatment for CSDH, the key questions are:

How to do it? Is it safe? Furthermore, does it work?

### 4.1. How to Do It?

The catheterization of the MMA ipsilaterally to the CSDH allows only minor technical variations. The penetration of the embolic agent deep into the dependent vasculature of the MMA is considered crucial by most authors. The following embolic agents have been used and were reported together with good results:-Polyvinyl alcohol particles (PVA) [2,28,29,37,38]-Embospheres [39,40]-n-butyl cyanoacrylate (nBCA) [25,36,39,41]-PVA and nBCA [42,43]-Absolute alcohol [44]-Onyx [35,45,46]-Phil: Pending-Squid: Pending-Coils [37,47].

Catapano et al. (2020) compared, in a small series (*N* = 35), microspheres, n-butyl cyanoacrylate, and Onyx as embolic agents and did not find differences concerning safety and efficacy margins [48]. Other authors came to the same conclusion [49].

The timing of follow-up NCCT is a matter of controversy. NCCT examination schedules at 1 day, 3 weeks, and 3 months, and later if required, appear reasonable [43].

### 4.2. Is It Safe to Embolize the MMA as a Treatment for CSDH?

Essentially yes. The large majority of authors encountered no or infrequent complications [19,36,40,50]. 

*Complications:* Access artery occlusion is rare but remains a concern [22]. Kan et al. (2020) evaluated a multicenter series of 154 MMA embolizations. They encountered one asymptomatic MMA rupture, one seizure, and one facial droop. Two patients experienced a worsening of their CSDH, which eventually turned fatal [49]. Martinez-Perez et al. (2020) reviewed 6 publications with 164 patients and calculated a complications rate for the embolization of 6% [51]. Jumah et al. (2020) evaluated 11 studies with 177 patients and calculated a 1.2% risk of embolization-related complications [52]. Joyce et al. (2020) evaluated 121 elderly patients with CSDH and MMA embolization. Adverse events were equally rare in the age groups 65–79 years (2.3%: 1× seizure, 1× infarction) and ≥80 years (1.6%: 1× temporary aphasia) [53].

Rather an interesting phenomenon than a significant complication is the *de novo* occurrence of a DAVF several months after the particle embolization of both MMAs [22]. The “DAVFs” described by Piergallini et al. (2019) occurred immediately after the particle embolization through large microcatheters (Headway 27, MicroVention). Their angiographic appearance is different from a “natural” DAVF, and they could result from periprocedural dissections or ruptures of dural arteries [54].

Raviskanthan et al. (2021) described bilateral abducens nerve palsies after bilateral MMA embolization with Embospheres [55]. 

Many patients presenting with CSDH are under medicinal anticoagulation or antiaggregation. While the burr-hole evacuation of a CSDH needs an interruption of the anticoagulation, embolization of the MMA can be carried out under these medications [37]. This can be viewed as an additional safety feature.

### 4.3. Does the MMA Embolization Work, and Will the CSDH Disappear?

Most likely, yes. MMA embolization was initially (and is still) used in patients after failed surgery [42]. In the meantime, it can be considered as a viable alternative to burr hole trephination [42]. 

(Complete) resolution: Embolization of the MMA is associated with a greater extent of CSDH volume reduction. The rate of (near) complete resolution of CSDH after MMA embolization is in the range of 30–55% [2,25,35,37,40,48,53]. Gomez-Paz et al. (2020) were able to show that upfront MMA embolization in CSDH with a midline shift is feasible. In their series of 23 patients, improvement was achieved within 2–4 weeks, and resolution occurred within less than 2 months [56].

Recurrence: In an early literature review from 2014, Chihara et al. reported on 45 published cases of MMA embolization for CSDH treatment with five recurrences. For their own case of CSDH recurrence after embolization, they discuss the role of an organized hematoma as a cause of delayed recurrence. This patient was treated with craniotomy and capsulectomy [57]. Saito et al. (2019) assume that angioneogenesis of the inner membrane of the CSDH is a reason for the recurrence of hematomas after MMA embolization [58].

Wang et al. (2020) discussed the pathomechanisms of recurrence in bilateral CSDH. They assume that recurrence in unilateral CSDH is mainly due to rebleeding from vessels supplied by the MMA. However, the recurrence of bilateral CSDHs on the untreated side is likely similar to the initial process of CSDH formation [59].

Several authors reported no recurrence after embolization [36,39,51].

Martinez-Perez et al. (2020) found a recurrence rate of 6.7% in their literature review, which is in line with the 6.5% reported by Kan et al. (2020) [49].

Haldrup et al. (2020) reviewed 18 publications reporting 191 patients. They calculated a recurrence rate after MMA embolization for primary and recurrent CSDH of 4.1% and 2.4%, respectively [60].

In the Link et al. (2019) series, 8.9% of the embolized patients needed additional surgery, while in 91.1%, the volume of the CSDH was stable or decreased, thus avoiding surgery [50]. A meta-analysis by Dian et al. (2021) calculated a 20% reduced risk of CSDH recurrence after MMA embolization compared to surgical evacuation [61]. 

In several studies and meta-analyses, the recurrence rate was higher in operated than embolized patients [19,22,52,62]. In a study by Schwarz et al. (2021), the combined surgery and subsequent embolization resulted in a need for a second operation in only 4.5% [30]. Tiwari et al. (2021) performed volume calculations of the CSDH after embolization and found an exponential volume decay [40].

Clinical improvement: A potential benefit of embolization was, in most studies, not necessarily reflected by the clinical outcome (i.e., mRS score) [63]. The rate of a poor clinical outcome (mRS > 2) in the embolization group (12.5%) and the conventional group (9.1%) was not significantly different in the meta-analysis by Srivatsan et al. (2019) [62].

Brain re-expansion and the rate of hematoma cure have been used instead to describe the value of MMA embolization [38]. In the sizeable multicenter series of Kan et al. (2020), 44/138 patients had improvement, and 11/138 had deteriorated in their mRS [49].

Clinical trials: Further evidence concerning the efficacy of MMA embolization and other therapeutic concepts for CSDH can be expected from RCTs [64]. Several of them are underway [65,66,67], and some are designed to evaluate specific aspects like embolization after surgical CSDH evacuation [68] or the use of certain embolic materials like PVA [69].

In the trial “Embolization of the Middle Meningeal Artery With ONYX™ Liquid Embolic System for Subacute and Chronic Subdural Hematoma (EMBOLISM),” Onyx (Medtronic) will be used as an embolic agent for the MMA. A total of 600 patients will be randomized into four arms: Surgical hematoma evacuation, surgical hematoma evacuation plus embolization, observation-only, and embolization only [70]. The expected study end date is April 2023.

The “SQUID Trial for the Embolization of the Middle Meningeal Artery for Treatment of Chronic Subdural Hematoma (STEM)” evaluates MMA embolization with Squid compared to burr-hole hematoma evacuation and medical management. A total of 228 patients will be enrolled, with an expected study end date of December 2021 [71].

## 5. Conclusions

Embolization of the MMA with Squid 12 and 18 can be used as an effective treatment for recurrent CSDH after surgical evacuation. Distal penetration is required through the shell vessels, as well as obliteration of all anastomoses on the opposite side and distribution in the newly formed vessels. The embolization causes damage to the CSDH “environment.” Embolization of the MMA also appears as a promising treatment in patients who previously were not operated on with mass-effect and a neurological deficit from a CSDH, who are otherwise considered candidates for surgical evacuation of the hematoma. Embolization is less invasive than open surgery. This is particularly relevant for patients with CSDH with comorbidities that require permanent anticoagulation or antiaggregation, which both are not ideal conditions for open microsurgery.

## Figures and Tables

**Figure 1 jcm-10-04436-f001:**
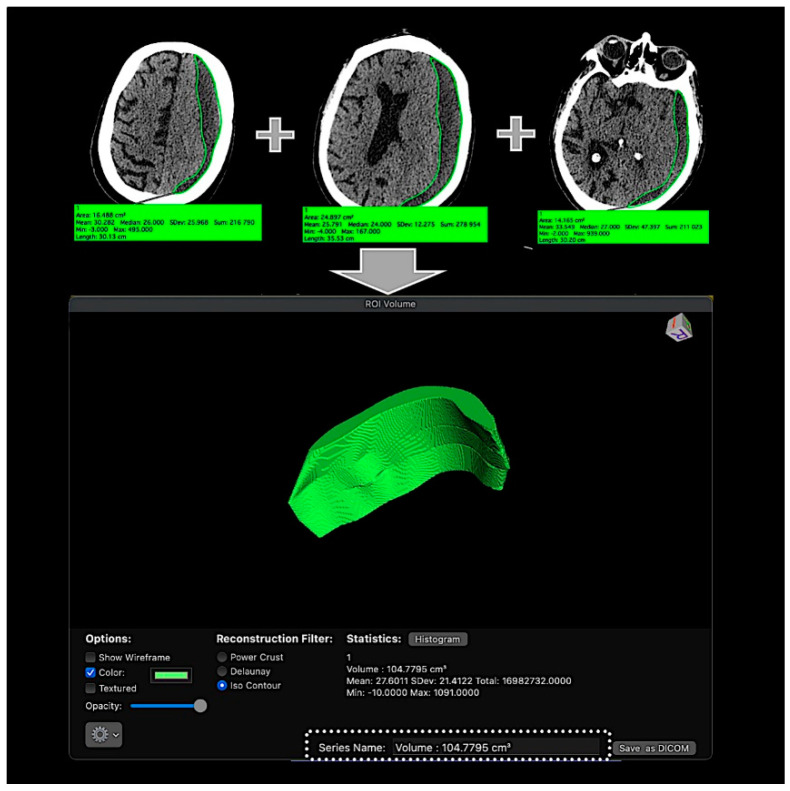
Volumetric analysis of a CSDH for a volume measurement using the OsiriX software. ROI segmentation was performed on each slice of an NCCT, and a volume-rendered model was produced for each scan, with an automated calculation of the entire volume.

**Figure 2 jcm-10-04436-f002:**
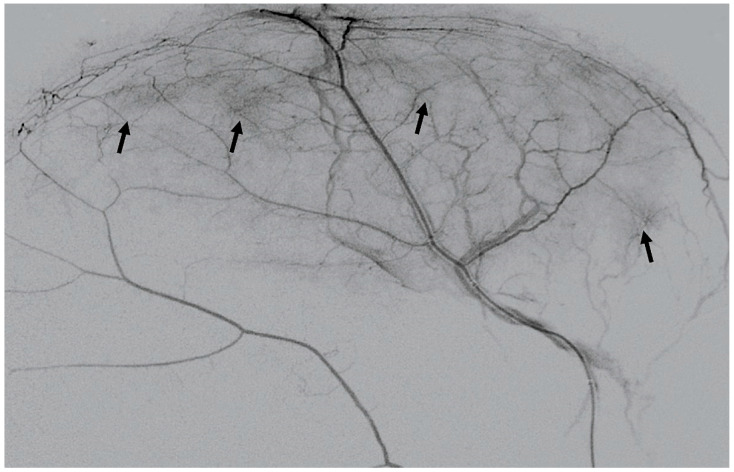
DSA, lateral view of a selective contrast medium injection of the MMA. Note the typical hypervascular areas (arrows).

**Figure 3 jcm-10-04436-f003:**
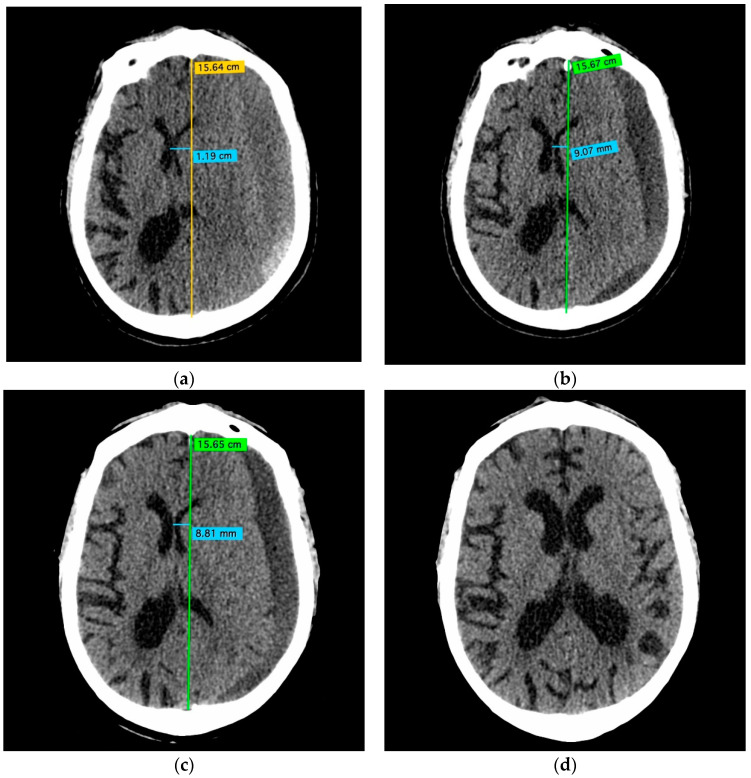
NCCT before (**a**), 5 days (**b**), 8 days (**c**), and 6 months (**d**) after the treatment of a left-sided CSDH with Squid embolization of the left MMA. The volume of the CSDH and the resulting midline shift were partially resolved within the first week after the embolization (**b**,**c**). Follow-up NCCT after 6 months (**d**) confirmed the complete resolution of the CSDH.

**Figure 4 jcm-10-04436-f004:**
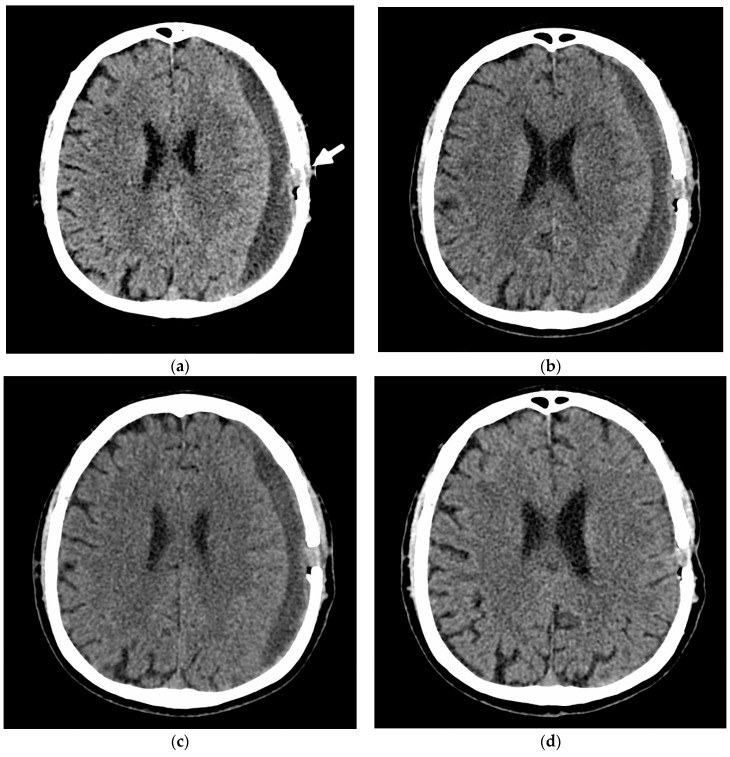
NCCT 28 days after surgical evacuation of a CSDH with recurrence and before embolization of the ipsilateral MMA (**a**). Four days (**b**) and 8 days (**c)** after the embolization, the CSDH is already seen to be shrinking. Four months after the embolization (**d**), the CSDH has entirely been resolved.

**Figure 5 jcm-10-04436-f005:**
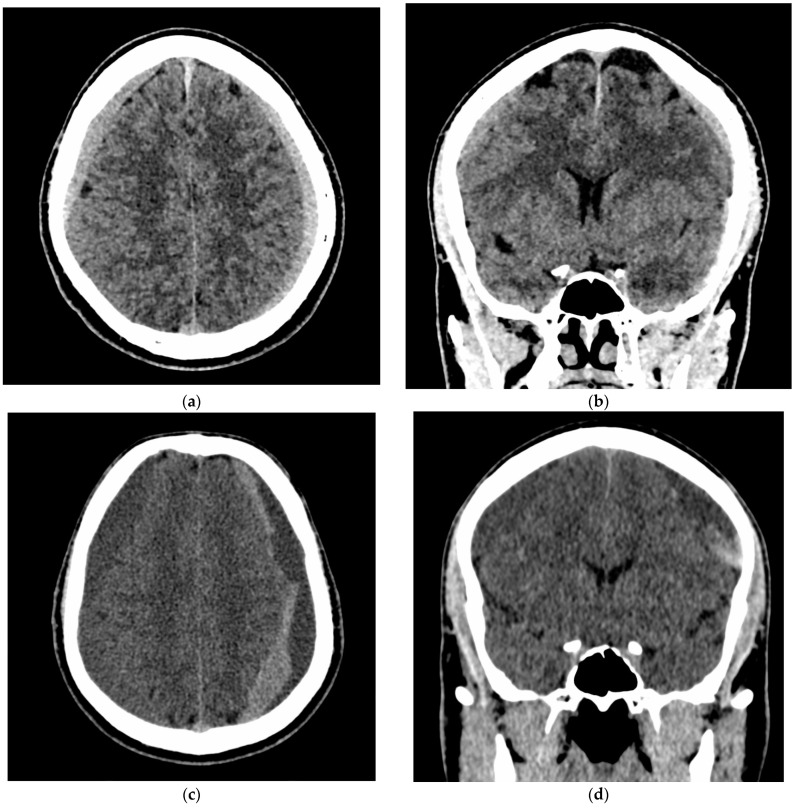
NCCT 30 days after closed head trauma and 1 month before the embolization shows bilateral CSDH (axial (**a**) and coronal (**b**)) images, calculated CSDH volumes: Left 68 mL, right 54 mL. The corresponding NCCT 1 month later and 1 day before the MMA embolization shows an increase of the CSDH volumes (axial (**c**) and coronal (**d**)) images, calculated CSDH volumes: left 83 mL, right 69 mL. Both MMAs were embolized with Squid. Posterior–anterior (**e**) and lateral (**f**) radiographic images show the radiopaque cast in both MMAs (arrows). An early follow-up NCCT on day 1 after the embolization ruled out any adverse event (axial (**g**) and coronal (**h**) scans). The volumes were calculated as 71 mL on the left and 68 mL on the right. One week after the embolization, the left CSDH was slightly larger than before, while the right CSDH was smaller by a volume of 13 mL (axial (**i**) and coronal (**j**) scans). One month after the embolization, both CSDHs were smaller, with a volume of 42 mL on the left and 47 mL on the right (axial (**k**) and coronal (**l**) scans). The final follow-up NCCT at 6 months demonstrated only minimal remnants of both CSDH (axial (**m**) and coronal (**n**) scans).

**Table 1 jcm-10-04436-t001:** Summary of demographic and treatment data of 10 patients with CSDH treated with Squid embolization of the MMA.

Patient # Age (Years) Gender	Side Symptoms	VolumePre Embol (mL)	Days Post Embol	Volume Post Embol (mL)	Surgery	Complication Outcome
1, 44, f	bilateral headache, nausea	right 18 left 9	2	right 16	pre embol	none mRS 0
				left 6		
			15	right 9		
				left 5		
			64	0		
2, 56, m	bilateral headache, nausea	right 22 left 25	1	right 21	none	none mRS 0
				left 23		
			5	right 21		
				left 38		
			57	0		
3, 42, m	bilateral headache, nausea	right 69 left 83	1	right 68	none	none mRS 0
				left 71		
			8	right 55		
				left 74		
			29	right 47		
				left 42		
			180	0		
4, 74, m	left headache, nausea	88	4	63	pre embol	none mRS 0
			8	35		
			120	0		
5, 80, m	bilateral headache, nausea	right 44 left 54	1	right 39	pre embol (right side)	none mRS 0
				left 44		
			12	right 33		
				left 29		
			16	right 28		
				left 25		
			45	right 15		
				left 19		
			210	0		
6, 70, m	left mild right-handed hemiparesis up to 4 points, seizures	105	1	96	none	none mRS 1
			6	85		
			10	79		
			17	76		
			180	0		
7, 89, f	bilateral tetraparesis (muscle strength up to 1–2 points on the right, up to 3 points on the left, depression of consciousness, obtundation)	right 55 left 67	1	right 53	none	none mRS 0
				left 66		
			3	right 49		
				left 62		
			23	right 38		
				left 56		
			31	right 25		
				left 32		
			90	0		
8, 75, m	left mild right-handed hemiparesis, aphasia, headaches, nausea	169	1	155	pre embol	none mRS 0
			5	142		
			8	138		
			60	0		
9, 59, f	right headache, nausea	65	1	47	pre embol	none mRS 0
			5	33		
			90	0		
10, 71, m	left headache, nausea	77	2	68	none	none mRS 0
			60	0		

## Data Availability

The data presented in this study are available on request from the first author. The data are not publicly available due to patient privacy protection.

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
