# Peer review of "Endovascular Treatment of Chronic Subdural Hematomas through Embolization: A Pilot Study with a Non-Adhesive Liquid Embolic Agent of Minimal Viscosity (Squid)"

_jcm, 2021, doi:10.3390/jcm10194436_

Round 1

Reviewer 1 Report

very well written mss re the embolization of SDH.

Author Response

We will submit the manuscript to the language revision service by MDPI.

Reviewer 2 Report

This manuscript concerns an area within CSDH management than are gaining increasing scientific interest.

The authors are testing the use of non-adhesive liquid embolic agent (squids) in endovascular embolization of the medial meningeal artery as a treatment in patients with CSDH. The outcomes are assessment of safety and efficiency.

In overall the authors have done a good job testing these squids in the MMA embolization. This is a well written manuscript in which the conclusion is well supported by the study. However, some areas should be addressed before I can endorse publication.

Major revisions:

  • Regarding the efficacy endpoint, the authors demonstrates a volume reduction in all patients and have done control CT after 45 to 180 days. I cannot see that the authors have performed a CT-angiography to examine if the MMA is truly occluded. CSDH are known to disappear spontaneously (Santarius T et al. Working toward rational and evidence-based treatment of CSDH. Clin Neurosurg. 2010;57:112-22. And Voelker JL. Nonoperative treatment of CSDH. Neurosurg Clin N Am. 200 Jul; 11(3):507-13). As such, the volume reduction could be a spontaneously resorption, if the MMA is still open following the endovascular embolization. Accordingly, some kind of demonstration of an actually closure of the MMA is needed. If this is not possibly, it should be acknowledge as a major limitation of the study.

Minor revisions:

  • The terms control CT, NCCT (which should be spelled out at the first appearance) and follow-up NCCT are used at random and should be standardized throughout the manuscript.
  • In the exclusion criteria symptoms regarding “clinical signs of acutely increased ICP” should be specified as the authors include patients with headache.
  • The authors discuss the timing and frequency of control CT postoperatively. I have difficulties reading if these control CT is performed because of the study or in general in CSDH patients. It should be acknowledged that standard use of control CT has been shown to results in overtreatment in CSDH patients surgically treated and this would properly also be the case for CSDH patients with MMA embolization (Schucht P, Fischer U, Fung C, Bernasconi C, Fichtner J, Vulcu S, et al. Follow-up Computed Tomography after Evacuation of Chronic Subdural Hematoma. N Engl J Med. 2019;380(12):1186-7.).
  • The authors mention histological examination of the CSDH membrane, but I have difficulties to understand if histological examination was done in this study or if it is a general part of the discussion. If it is a part of this study, it is not within the scope of the manuscript and, in my opinion, should be left out. It is confusing for the reader.
  • When did the 3 patients with neurological deficits experience symptom remission? If open burrhole surgery provides an instant pressure relief and symptom remission, this could still be superior to MMA embolization alone in which pressure relief may be slower leading to longer symptom duration. This could also be acknowledge in a limitation section.

Author Response

Major revisions:

>>Regarding the efficacy endpoint, the authors demonstrates a volume reduction in all patients and have done control CT after 45 to 180 days. I cannot see that the authors have performed a CT-angiography to examine if the MMA is truly occluded. CSDH are known to disappear spontaneously (Santarius T et al. Working toward rational and evidence-based treatment of CSDH. Clin Neurosurg. 2010;57:112-22. And Voelker JL. Nonoperative treatment of CSDH. Neurosurg Clin N Am. 200 Jul; 11(3):507-13). As such, the volume reduction could be a spontaneously resorption, if the MMA is still open following the endovascular embolization. Accordingly, some kind of demonstration of an actually closure of the MMA is needed. If this is not possibly, it should be acknowledge as a major limitation of the study.

 << The following has been added: Under 2.4. “DSA follow-up between month 3 and month 6 after the embolisation has been offered to all patients.

Under 3.5 So far 7 DSA examinations have been carried out and they confirmed the permanent occlusion of all embolized MMAs.

Minor revisions:

>>The terms control CT, NCCT (which should be spelled out at the first appearance) and follow-up NCCT are used at random and should be standardized throughout the manuscript.

<< The following has been added or changed: NCCT has been used throughout and has been explained on the occasion of its first appearance

>>In the exclusion criteria symptoms regarding “clinical signs of acutely increased ICP” should be specified as the authors include patients with headache

<<The following has been added:  Exclusion criteria were: symptomatic, with clinical signs of acutely increased ICP, e.g., impaired consciousness, vomiting without nausea, papilledema

 >>The authors discuss the timing and frequency of control CT postoperatively. I have difficulties reading if these control CT is performed because of the study or in general in CSDH patients. It should be acknowledged that standard use of control CT has been shown to results in overtreatment in CSDH patients surgically treated and this would properly also be the case for CSDH patients with MMA embolization (Schucht P, Fischer U, Fung C, Bernasconi C, Fichtner J, Vulcu S, et al. Follow-up Computed Tomography after Evacuation of Chronic Subdural Hematoma. N Engl J Med. 2019;380(12):1186-7.). 

<< The follow-up computed tomography was done 1 day, 1 week, 1 month, 3 months and 6 months after embolisation and was part of the study in order to rule out or recognize possible complications of the embolization treatment.

<<< Commentary: There was no risk of overtreatment  since all CSDH had resolved at the time of follow-up treatment. This was rather complete treatment and not overtreatment.

>>The authors mention histological examination of the CSDH membrane, but I have difficulties to understand if histological examination was done in this study or if it is a general part of the discussion. If it is a part of this study, it is not within the scope of the manuscript and, in my opinion, should be left out. It is confusing for the reader.

<< The following has been changed in the text: “Histological results of the CSDH multilayer membrane, published by other groups,  revealed”

<<<Commentary: We have no histological results so far and we mention this aspect as a general part of the discussion. As such the histological changes reported in the literature and cited by us are crucial for the understanding of the underlying pathophysiology. We therefore propose to maintain the actual form. 

>>When did the 3 patients with neurological deficits experience symptom remission? If open burr-hole surgery provides an instant pressure relief and symptom remission, this could still be superior to MMA embolization alone in which pressure relief may be slower leading to longer symptom duration. This could also be acknowledge in a limitation section.

<<The following has been added under 3.2 Patient population: “All three patients improved clinically within the first week after the embolization.”

Reviewer 3 Report

Review "Endovascular treatment of chronic subdural hematomas through embolization with a non-adhesive liquid embolic agent of minimal viscosity (Squid)".

The authors present a single-center, prospective observational study of 10 patients treated on chronic subdural hematomas by endovascular embolization of the middle meningeal artery.

The study was conducted to evaluate the safety and effectiveness of the use of liquid embolic agent of minimal viscosity (Squid) in endovascular treatment of these patients. Primary endpoint was the rate of procedural complications, secondary endpoint a 50% or more reduction of cSDH volume in follow-up CCT.

Of the 10 patients, 5 had bilateral cSDH and 5 had unilateral cSDH. Previous surgical removal of hematoma in 5 patients had no effect on recurrent/residual subdural blood volume.

After endovascular treatment of the MMA ipsilateral to the cSDH, safety endpoints were met without any adverse events. The secondary endpoint was archieved in 10/10 patients. They conclude that is is a safe and efficient alternative procedure to treat patients with cSDH.

This is a very interesting paper. I have some questions for further improvement: The inclusion/exclusion criteria are multifold and have some issues to be adressed: - two different groups were intermingled: according to the criteria on page 3, patients with or without previous surgery for cSDH could be included - The definition of the A minimum time span between surgery and MMA embolization was neither given nor mentioned - Why would you treat an asymptomatic patient with a small cSDH?

More issues - How many patients between 11/2019 and 7/2021 did you have to screen? - How high was the drop-out rate of screened and included patients? - Did all patients receive a follow-up investigation at 1 day, 1 week, 1 and 3 months after treatment? On page 8 you only stated that „All patients underwent at least one follow-up NCCT examination“. The methods of SDH volume measurement and the endovascular procedure are described well and in detail. Is there a cut-off value for the number of minimum embolizations to achieve the secondary endpoint?

How did you make sure that after 3 months, the MMA is still closed by Squid?

With your excellent work-up, this procedure can be conducted with more patients to underline your preliminary results.

Thus, I would suggest to add the words „preliminary results“ or „pilot study“ or „case series“ in the title of your work. In my opinion, this manuscript qualifies for publication after adressing above mentioned issues.

Author Response

>>The inclusion/exclusion criteria are multifold and have some issues to be adressed: - two different groups were intermingled: according to the criteria on page 3, patients with or without previous surgery for cSDH could be included - The definition of the A minimum time span between surgery and MMA embolization was neither given nor mentioned - Why would you treat an asymptomatic patient with a small cSDH?

<<The following was added under inclusions criteria: “no defined time interval between surgery and embolization was applied; a minimum depth of 10 millimeters was considered necessary to justify embolisation” 

>>More issues - How many patients between 11/2019 and 7/2021 did you have to screen? How high was the drop-out rate of screened and included patients?

<< The following has been added under 3.2 Patient population: “Between November 2019 and July 2021, endovascular treatment was offered to 12 patients and accepted by 10 patients (7 male) with CSDH and following the study's inclusion and exclusion criteria.“

<<< Accordingly the dropout rate was 16,6 %

>>Did all patients receive a follow-up investigation at 1 day, 1 week, 1 and 3 months after treatment?

<< Yes, all the patients were followed-up at 1 day, 1 week, 1 and 3 months after embolization.

>>On page 8 you only stated that „All patients underwent at least one follow-up NCCT examination“. The methods of SDH volume measurement and the endovascular procedure are described well and in detail. Is there a cut-off value for the number of minimum embolizations to achieve the secondary endpoint?

<< The following was added:  “direct embolization of the distal MMA branches in a single session”

>>How did you make sure that after 3 months, the MMA is still closed by Squid?

  << The following has been added: Under 2.4. “DSA follow-up between month 3 and month 6 after the embolisation has been offered to all patients.

Under 3.5 So far 7 DSA examinations have been carried out and they confirmed the permanent occlusion of all embolized MMAs.

>>With your excellent work-up, this procedure can be conducted with more patients to underline your preliminary results.Thus, I would suggest to add the words „preliminary results“ or „pilot study“ or „case series“ in the title of your work. In my opinion, this manuscript qualifies for publication after adressing above mentioned issues.

<< The title has been changed as proposed: “Endovascular Treatment of Chronic Subdural Hematomas Through Embolization: A pilot study with a Non-adhesive Liquid Embolic Agent of Minimal Viscosity (Squid)”

Reviewer 4 Report

This is a well written paper about a small study of CSDH treatment with Squid12 and 18 injection into the middle meningeal artery, illustrating the effectiveness of the treatment.

It´s main weakness is obviously the small patient size. Since the study design seems to be prospective, it is not absolutely clear, why the authors did not plan with a higher patient number. Perhaps the authors could detail a bit more on the study design.

Overall I feel that the discussion, although being very comprehensive, could be shortened.

Some specifics:

P3| “Inclusion and Exclusion Criteria…” the exclusion and inclusion criteria lists both start with the same bullet point “Patient with CSDH, confirmed by NCCT or MRI…”, this is confusing

P4| “2.3 Endovascular Treatment….” The bullet points got mixed up, which makes the list hard to understand.

P5| “the anterior…” I think there is some grammatical error in this sentence

P5| “on average, one vial….” This passage is somehow contradictionary, since one vial of Squid has 1,5ml of volume, which means that injection of 2,1 ml using one vial is not possible. Perhaps this passage could be combined with the description of the injection technique (Squid12 > 14 > 12) to make the injection technique clearer to the reader

P6| “Squid…” do I understand correctly, they injected first Squid12, then Squid18 and then Squid12 again to obliterate the vessels?

P6| “the proximal segment of….” It would be nice, if the authors would give a specific anatomic landmark or show a picture, that illustrates what “proximal part” means to them, thus making it clearer which parts of the MMA should be avoided.

Author Response

>>P3| “Inclusion and Exclusion Criteria…” the exclusion and inclusion criteria lists both start with the same bullet point “Patient with CSDH, confirmed by NCCT or MRI…”, this is confusing

<< The first sentence under exclusion criteria has been changed as follows: “symptomatic CSDH, confirmed by NCCT or MRI, with clinical signs of acutely increased ICP (e.g., impaired consciousness, vomiting without nausea, papilledema)”

>>P4| “2.3 Endovascular Treatment….” The bullet points got mixed up, which makes the list hard to understand.

<< This occurred during submission and has been corrected as follows: ”The strategy of the endovascular treatment of CSDH was oriented towards the following goals:

-           direct embolization of the distal MMA branches in a single session

-           avoidance of reflux into the proximal segments of the MMA in order to avoid the dissemination of Squid through dangerous anastomoses

-           termination of blood supply to the vessels of the CSDH capsule both due to occlusion of the meningeal vessels, and due to embolization of the vessels of the capsule of the CSDH by non-adhesive embolizing material of low viscosity to prevent recurrence and increase of the hematoma volume

-           penetration of the non-adhesive embolizing agent through the collaterals to the distal branches of the opposite MMA, preventing the blood supply of the CSDH capsule from the opposite side

-           acceleration of the processes of hematoma absorption and decompression of adjacent brain”

>>P5| “the anterior…” I think there is some grammatical error in this sentence

<<< The entire text has undergone extensive language revision.

>>P5| “on average, one vial….” This passage is somehow contradictionary, since one vial of Squid has 1,5ml of volume, which means that injection of 2,1 ml using one vial is not possible. Perhaps this passage could be combined with the description of the injection technique (Squid12 > 14 > 12) to make the injection technique clearer to the reader

<< The following has been added: “On average, one vial of Squid 12 and one or two vials of Squid 18 per side was used.”

>>P6| “Squid…” do I understand correctly, they injected first Squid12, then Squid18 and then Squid12 again to obliterate the vessels?

<< Yes, this is correct.

>>P6| “the proximal segment of….” It would be nice, if the authors would give a specific anatomic landmark or show a picture, that illustrates what “proximal part” means to them, thus making it clearer which parts of the MMA should be avoided.

<< The following has been added: “Embolisation should be limited to the frontal and parietal branches of the MMA, avoiding the ophthalmic artery anastomosis, the petrosal branch and the foraminal segment”.